# Unravelling Anti-Melanogenic Potency of Edible Mushrooms *Laetiporus sulphureus* and *Agaricus silvaticus* In Vivo Using the Zebrafish Model

**DOI:** 10.3390/jof7100834

**Published:** 2021-10-05

**Authors:** Aleksandar Pavic, Tatjana Ilic-Tomic, Jasmina Glamočlija

**Affiliations:** 1Institute of Molecular Genetics and Genetic Engineering, University of Belgrade, Vojvode Stepe 444a, 11042 Belgrade, Serbia; tatjanait@imgge.bg.ac.rs; 2Institute for Biological Research “Siniša Stanković”, University of Belgrade, Bulevar Despota Stefana 142, 11060 Belgrade, Serbia

**Keywords:** inhibition of melanogenesis, *Laetiporus sulphurous*, *Agaricus silvaticus*, zebrafish toxicity, inflammation, neutropenia, tyrosinase, melanin, kojic acid, hydroquinone

## Abstract

Severe drawbacks associated with the topical use of depigmenting agents in treatments of skin hyperigmentations impose a great demand for novel, effective, and safe melanogenesis inhibitors. Edible and medicinal mushrooms, known for numerous health-promoting properties, represent a rich reservoir of anti-melanogenic compounds, with the potential to be applied in preventing excessive skin pigmentation. Herein, using zebrafish (*Danio rerio*) as a preclinical animal model, we have demonstrated that ethanol extract of *Laetiporus sulphureus* (LSE) and *Agaricus silvaticus* (ASE) are not toxic at high doses up to 400–500 µg/mL while effectively inhibit melanogenesis in a dose-dependent manner. At depigmenting doses, the explored extracts showed no adverse effects on zebrafish embryos melanocytes. Even more, they did not provoke inflammation or neutropenia when applied at the highest dose ensuring almost complete the cells depigmentation. Since LSE and ASE have demonstrated significantly higher the therapeutic potential than kojic acid and hydroquinone, two well-known depigmenting agents, overall results of this study strongly suggest that the explored mushrooms extracts could be used as efficient and safe topical agents in treatments of skin hyperpigmentation disorders.

## 1. Introduction

In mammals, the color of skin and hair is attributed by presence of the pigment melanin, a biopolymer that is crucial for protecting the skin from harmful ultraviolet radiation and preventing cancer. However, its excessive accumulation in melanocytes leads to various hyperpigmentation skin disorders such as melasma, freckles, lentigines, chloasma, ephelides and melanoderma [1]. The visible nature of these dermatological diseases results in a significant psychological impact on patients, especially because skin diseases have a strong impact on the physical appearance and emotional well-being [2]. Therefore, hampering melanin production and preventing its accumulation in melanocytes is the major strategy in current dermatology and cosmetics aiming to prevent hyperpigmentations without the effect on melanocytes viability [3].

Since the tyrosinase, a copper-containing enzyme, plays a key role in the melanin biosynthesis, inhibition of its activity by tyrosinase inhibitors is an essential endeavor in treatments of hyperpigmentation disorders, and an important strategy in the developmental pipeline for novel depigmenting drugs for further application in skin pharmacology and cosmetics [4,5,6,7]. Unfortunately, many tyrosinase inhibitors known so far, such as kojic acid, hydroquinone, azelaic acid, arbutin, and phenylthiourea, are documented to possess numerous drawbacks, including poor skin penetration, low stability, irreversible melanocyte loss, as well as mutagenic, clastogenic, and even cancerogenic potential [6], hindering their long-term use. For these reasons, there is still a great demand in the skin pharmacotherapy for novel, effective, safe, and stabile tyrosinase-inhibiting agents which could be applied in the treatment of dermatological disorders [8].

Medicinal and edible fungi are well documented for their health-promoting activities in humans [9,10], being used as food sources in traditional folk medicine for thousands of years in many different countries. Their fruiting bodies (mushrooms) containing various biologically active metabolites represent a rich resource of antiviral, antivirulence, antimicrobial, anticancer, anti-diabetes, immunostimulatory, and anti-obesity molecules. In addition, mushrooms present a natural untapped reservoir of depigmenting molecules [11,12], the potential of which is still weakly explored.

As a critical role in melanin production is determined by tyrosinase, the traditional approach in a screen for melanogenesis inhibitors is based on the use of purified mushroom tyrosinase (*Agaricus bisporus* enzyme) and an assessment of the inhibitory effects on enzyme activity. However, it has clearly been proven that many melanogenesis inhibitory agents active in this in vitro model simply failed to decrease pigmentation in human melanocytes and vice versa. The main reason for such discrepancy lies in the crucial differences in the structure, cellular localization, and the substrate’s affinity between mushroom and human tyrosinase [6]. One such example is the activity of thiamidol, a resorcinol-derived compound, which demonstrated potent activity on human tyrosinase and prominent efficacy in treatment of various skin hyperpigmentations in human subjects, while being inactive against mushroom tyrosinase [13,14]. In recent years, the zebrafish (*Danio rerio*) has emerged as a leading preclinical animal model in the pharmacological search for novel, effective, and safe depigmenting agents owing to their a high correlation with humans in molecular biology of melanogenesis [15,16,17]. Similarly to hormonal regulation of melanin pigmentation in mammalian skin including humans [2], the melanogenesis in zebrafish skin is a hormonally regulated process which includes α-melanocyte-stimulating hormone (α-MSH) and melanin-concentrating hormone (MCH) action [18]. However, the main strength of the zebrafish model relies on almost total optical transparency of the developing embryos that provides a unique opportunity to easily identify depigmenting agents without adverse effects on skin melanocytes and evaluate their efficacy related to hyperpigmentation disorders.

Bearing in mind an exceptional medicinal importance of macrofungi, Glamoclija and collaborators conducted the extensive studies on biological activities of edible and medicinal mushrooms, pointing them out as a rich resource of bioactive molecules with antibacterial, antifungal [19], antioxidative [20], antiangiogenic, and anticancer [21] properties. In this study, we have explored the depigmenting potential of the ethanol extract of five edible mushrooms (*Lethiporus sulphureus, Agaricus silvaticus, Agrocybe aegerita, Pleurotus ostreatus, Polyporus squamosus*) in vivo and comprehensively assessed their possibility to be applied as topical depigmenting agents.

## 2. Material and Methods

### 2.1. Collecting and Preparation of Ethanol Extracts of Fruiting Bodies

Fresh fruiting bodies of *Lethiporus sulphureus, Agaricus silvaticus*, and *Polyporus ostreatus* were harvested in forests from Tara Mountain, in western Serbia, in September 2020. *Agrocybe aegerita* and *Polyporus squamosus* were collected from popular trees at Jabučki rit (northern Serbia) during April 2020 and authenticated by Jasmina Glamočlija (Institute for Biological Research, the University of Belgrade, Serbia). Samples were prepared, as previously described Petrovic et al., 2020 [21]. A voucher specimen was deposited at the Fungal Collection Unit of the Mycological Laboratory, Department of Plant Physiology, Institute of Biological Research “Siniša Stanković”, Belgrade, the Republic of Serbia, under the number Ls-09-2020, Asyl-001-2020, Po-09-2020, Aa-04-2020 and Ps-04-2020. Ethanol extracts of the collected mushrooms were prepared following the procedure, as previously described Petrovic et al., 2020. The homogenized fruiting bodies (10 gr) were suspended in 340 mL of boiling distilled water (2 h). Obtained homogenate was filtered and pellet re-extracted two more times with 340 mL of boiling distilled water over 2 h. Supernatants were pooled, frozen at −20 °C, and lyophilized. The residue from hot aqueous extraction was extracted in 95% ethanol overnight at 4 °C. The extracts were then centrifuged at 3000 rpm for 15 min, filtered through the Whatman No 4 filter paper, and dried by a rotary evaporator (BuchiR-210, Flawil, Switzerland) under a vacuum at 40 °C.

### 2.2. In Vivo Toxicity Assessment in the Zebrafish Model

Toxicity evaluation of fungal ethanol extracts was carried in the zebrafish (*Danio rerio*) model, according to previously published procedure [21,22] following the general rules of the OECD Guidelines for the Testing of Chemicals (OECD, 2013, Test No. 236) [23]. All experiments involving zebrafish were performed in compliance with the European directive 2010/63/EU and the ethical guidelines of the Guide for Care and Use of Laboratory Animals of the Institute of Molecular Genetics and Genetic Engineering, University of Belgrade. Wild type (AB) zebrafish were kindly provided by Dr Ana Cvejić (Wellcome Trust Sanger Institute, Cambridge, UK), raised to adult stage in a temperature- and light-controlled zebrafish facility at 28 °C and standard 14:10-h light-dark photoperiod, and regularly fed with commercially dry food (SDS200/300 granular food; Special Diet Services, Essex; UK and TetraMinTM flakes; Tetra Melle, Germany) twice a day and *Artemia nauplii* once daily. Embryos produced by pair-wise mating were collected and distributed into 24-well plates containing 10 embryos per well and 1 mL of E3 medium (5 mM NaCl, 0.17 mM KCl, 0.33 mM CaCl_2_, and 0.33 mM MgSO_4_ in distilled water), and raised at 28 °C. In order to assess the dose-dependent lethality and developmental teratogenicity, the embryos at the 6 h post-fertilization (hpf) stage were treated with six different concentrations of the tested extracts (50, 100, 200, 300, 400, and 500 µg/mL), and inspected for apical endpoints (Appendix A) every day by 120 hpf upon an steremicroscope (Carl Zeiss™ Stemi 508 doc Stereomicroscope, Berlin, Germany). Kojic acid and hydroquinone, two marketed depigmenting agents, were used as positive controls. DMSO (0.25%) was used as a negative control. Experiments were performed three times using 20 embryos per concentration. Dead embryos were discarded every 24 h. At 120 hpf, alive embryos were inspected for the morphology, anesthetized with the addition of 0.1% (*w/v*) tricaine solution (Sigma-Aldrich, St. Louis, MO, USA), and then photographed and killed by freezing at −20 °C for ≥ 24 h.

### 2.3. Inflammatory and Immunosuppressive In Vivo Response Determination

To evaluate the extracts with depigmenting activity (ASE and LSE) for the possible inflammatory and/or immunosuppressive effects, 24-hpf-old embryos of transgenic *Tg*(*mpx*:GFP) i114 zebrafish line expressing GFP in neutrophils [24] were exposed to the two highest anti-melanogenic doses of each extract (300 i 400 µg/mL of ASE and 400 i 500 µg/mL of LSE). At 72 hpf, embryos were imaged under a fluorescence microscope (Olympus BX51, Applied Imaging Corp., San Jose, CA, USA), and the neutrophils occurrence (fluorescence intensity) was determined using the ImageJ program (NIH public domain software; NIH is National Institutes of Health). The experiment was performed three times using 10 embryos per a concentration. Kojic acid (500 i 2500 µg/mL) and hydroquinone (5 and 10 µg/mL) were used as the positive controls causing neutrophils depletion (neutropenia).

### 2.4. Determination of In Vivo Depigmenting Activity in the Zebrafish Model

Depigmenting activity was determined by measuring the melanin amount in zebrafish embryo after 48-h treatment according to the previously described procedure [8]. Briefly, after exposure for 48 h to the different doses of ASE, LSE, kojic acid, and hydroquinone, 100 embryos per concentration were sonicated in cold lysis buffer (20 mM sodium phosphate (pH 6.8), 1% Triton X-100) and then clarified by centrifugation at 13,000 rpm for 30 min at 4 °C (Eppendorf, 5315 D, Hamburg, Germany). The protein content in the lysates was determined with the Bradford assay using BSA as the standard. To determine relative melanin amount, the pellet was resuspended in 1 mL of 1M NaOH at 95 °C for 1 h, and the absorbance at 490 nm was measured using multiplate reader (InfiniteM200pro, Tecan, Männedorf, Switzerland). The specific melanin content was adjusted by protein amount in the same sample, and expressed as the percentage change in comparison to melanin in the control group, which was considered as 100%. Kojic acid was used as a positive control, known to decrease the melanin content.

### 2.5. Tyrosinase Inhibition Activity in Zebrafish Embryos

Tyrosinase activity was spectrophotometrically determined, as was previously described by Choi et al. [25]. Briefly, the protein amount (250 µg) from the lysate was added into the reaction mixture containing 50 mM of phosphate buffer (pH 6.8) and 2.5 mM of L-DOPA. The reaction mixture was incubated at 37 °C for 6 h and the absorbance at 475 nm was measured using a multireader (InfiniteM200pro, Tecan, Männedorf, Switzerland). The final activity was expressed as a percentage of the activity of the control. Kojic acid was used as a control.

## 3. Statistical Analysis

The experimental results were expressed as mean values ± SD. The differences in inhibitory effect on melanin synthesis and tyrosinase activity between the untreated and treated groups were determined according to the one-way ANOVA followed by a comparison of the means by the Bonferroni test (*p* = 0.05). All analyses were performed using an SPSS 20 (SPSS Inc., Chicago, IL, USA) software package.

## 4. Results and Discussion

Fungal extracts provide a chemical diversity of bioactive compounds (phenolics, terpenoids, polysaccharides, lectins, sterols, glycoproteins, and several lipids) with anti-melanogenic properties [26], many of which could realize the same activity by affecting diverse molecular targets and enzymes. In this sense, mushroom extracts could be regarded as topical agents in the treatment of skin hyperpigmentation disorders or explored as a rich reservoir of pure depigmenting molecules [27,28,29,30]. Moreover, they have become more prevalent as functional additives in commercial formulations owing to the numerous beneficial effects on skin health and heightened consumers’ concern regarding synthetic ingredients [30].

Accordingly, in this study, we selected the ethanol extract of five edible mushrooms (*Lethiporus sulphureus, Agaricus silvaticus, Agrocybe aegerita, Pleurotus ostreatus, Polyporus squamosus*) and evaluated them for a spectrum of bioactivities in vivo using the zebrafish model, including toxicity profiling and depigmenting activity. Zebrafish (*Danio rerio*) was proven as a reliable alternative to mammalian hosts due to their high genetic, molecular, physiological, and immunological similarity to humans, and high correlation in response to pharmaceuticals, including melanogenesis inhibitory molecules [31,32]. Nowadays, the zebrafish model represents a versatile biotechnological platform for new drug discovery and toxicity evaluation of natural and naturally derived bioactive compounds, simplifying the path to clinical trials and reducing the failure of potential therapeutics at later stages of testing [33].

Herein, we firstly addressed a dose-dependent toxicity of the selected fungal extracts exposing zebrafish embryos at the 6-hpf stage (an early developmental stage, ensuring the high sensitivity to the tested compounds) to six different doses over a period of 5 days. Besides embryos survival, we also assessed the teratogenicity, inner organs’ development and depigmenting phenotype of the LSE and ASE-treated embryos. The data obtained in this assay revealed different safety profiles of the explored extracts, ranking them in the toxicity according to the LC_50_ values as follow: *P. squamosus* > *A. silvaticus* > *P. ostreatus* > *L. sulphureus* > *A. aegerita* (Figure 1).

Out of the five extracts explored in vivo *A. aegerita* (AEE) followed by *L. sulphureus* (LSE) exhibited the best toxicological profiles, where the embryo exposed to extracts’ doses up to 600 µg/mL did not show any sign of toxicity, suggesting that AEA and LSE could be safe for human use. This was the case for *A. silvaticus* extract (PSE), which had doses up to 400 µg/mL. On the other side, *P. ostreatus* appeared to be very toxic, provoking multiple signs of cardiotoxicity already at the doses ≥ 200 µg/mL, such as pericardial edema, changes in heart morphology and a decline in heart contractility (Figure 1B).

Since an early exposure of zebrafish embryos to fungal extracts in the toxicity assay offers the unique possibility to identify those with depigmenting activity, the treated embryos were inspected for the body pigmentation at 72 hpf and the extracts with anti-melanogenic properties, and no side effects were selected for further examination. As shown in Figure 1, we found that only *Agaricus silvaticus* (ASE) and *Lethiporus suplhureus* (LSE) out of the five tested fungal extracts successfully inhibited embryos pigmentation. These extracts were comprehensively examined for the anti-melanogenic potency. Contrary, embryos treated with *Polyporus squamosus* extract (PSE) developed well pigmented but small or less stellate or even round melanocytes, indicating that PSE is melanocytotoxic (Figure 1B).

### 4.1. L. sulphureus and A. silvaticus Ethanol Extracts Possess Strong Depigmenting Potential In Vivo

Results on the good safety profile of ASE and LSE obtained in the previous in vivo assay prompted us to investigate the depigmenting potency of both fungal extracts and regard them for a possible use as topical agents in the treatment of skin hyperpigmentation disorders.

We firstly explored their dose-dependent melanogenesis inhibitory activity and compared them in the depigmenting efficacy with kojic acid and hydroquionone, two well-known melanogenesis inhibitors. The AB embryos were treated at the 22–23 hpf stage, when melanocytes were already developed, and analyzed for the body pigmentation. At the same time, the embryos were followed for the possible side effects of applied extracts on inner organ development. The kojic acid and hydroquinone were used as reference compounds, as the developmental toxicity in the zebrafish model has previously been demonstrated [8].

After the 2-day treatments (from 24 to 72 hpf), we found that both *A. silvaticus* and *L. sulphureus* extracts efficiently inhibited melanogenesis in vivo in a dose-dependent manner (Figure 2B), whereas LSE appeared to be the more effective one. In LSE-treated groups, embryos were less pigmented already at the dose of 50 µg/mL (data not shown), while at 100 µg/mL, the melanin amount was reduced by 71% in relation to that in the control (DMSO-treated) group (Figure 2C,D). In addition, the melanocytes pigmentation was completely inhibited at 500 µg/mL of LSE, whereas no adverse effects were observed in any treated fish. On the other side, when applied at the highest non-toxic dose of 400 µg/mL, ASE reduced melanin levels by 75–80%, as achieved upon 300 µg/mL of LSE, indicating that ASE has less anti-melanogenic potential than LSE.

It is important to emphasize that both fungal extracts demonstrated much higher anti-melanogenic potency than kojic acid, a marketed agent present in many skin whitening preparations (Table 1). The extracts were at up to 17 times more effective than kojic acid since the inhibitory effect on melanin synthesis (melanin amount) achieved at 300 µg/mL and 400 µg/mL of LSE and ASE, respectively, was only achieved at high kojic acid dose of 2500 µg/mL (Figure 2A). However, the extracts were less effective than hydroquinone, which reduced embryos pigmentation at 5 µg/mL and completely depigmented embryos at 10 µg/mL (Figure 2A).

However, in contrast to kojic acid and hydroquinone which caused toxic side effects already at doses below the effective (depigmenting) ones (Figure 2A), neither LSE or ASE elicited the toxic response at doses that almost completely prevented embryos pigmentation (Figure 2B). Moreover, bearing in mind that zebrafish embryos were challenged to high doses of ASE and LSE at the early developmental stage, a good therapeutic profile of both extracts and their possibility to be applied in treatments of skin hyperpigmentation disorders is clearly demonstrated, even during the pregnancy.

Nonetheless, it is of great importance to emphasize that the comprehensive evaluation of new depigmenting agents for teratogenic potential is a crucial step in the drug development process, since some skin-whitening agents, such as hydroquinone, are considered to possess detrimental effects on human embryo-fetal development [34,35]. In this study, we found that hydroquinone was much more teratogenic than kojic acid. During a course of 5-day exposure, this whitening agent weakly decreased embryos pigmentation but induced pericardial edema at the dose of 5 µg/mL, while the embryos depigmented at the doses ≥10 µg/mL were seriously arrested in the growth and suffered of severe cardiovascular issues (large pericardial edema and slow heart beating) and multiple skeletal malformations (microcephaly, scoliosis, malformed eyes, and otholiths) (Figure 2A). Similarly, the treatment with kojic acid caused an appearance of pericardial edema already at dose of 100 µg/mL with no effect on embryos melanogenesis, while embryos exposed to 2500 µg/mL of kojic acid were largely depigmented but showed severe cardiac toxicity and scoliosis. Taken together, these data additionally suggest that exposure to hydroquinone and kojic acid could present a high risk for embryo development if drugs are applied during pregnancy, leading to various teratogenic malformations and embryo loss. This is of particular matter if we consider that 35–45% of topically applied hydroquinone passes through the skin and goes directly into the bloodstream [36], while the successful treatment of certain skin hyperpigmentations, such as melasma, requires several months of drug application [37].

Nevertheless, although hydroquinone was used for decades in dermatology as the tyrosinase inhibitors for the hyperpigmentation treatments, such as lentigines, melasma, and freckles, the clinical use of this remedy is forbidden in the EU, USA, and many other countries from 2001 onwards due to its numerous side effects including exogenous ochronosis, melanosome damage, permanent melanocyte loss, local neutropenia, bone marrow toxicity, mutagenicity, etc. [38]. Furthermore, many other commercial skin-whitening compounds also possess various drawbacks, such as insolubility and poor bioavailability (ellagic acid), chemical instability, and bone marrow toxicity (arbutin) [39], as well as even carcinogenicity, poor skin penetration, and weak efficacy (kojic acid) [6]. Therefore, these data undoubtedly highlight that discovery of safe depigmenting compounds is of a pivotal significance.

### 4.2. LSE and ASE Are Not Melanocytotoxic to Zebrafish Embryos

Since the topical application of some depigmenting agents causes the irreversible skin melanocytes loss due to melanocytotoxic activity, leading to the development of new skin disorders and limited use (e.g., hydroquinone-based products), we further examined LSE and ASE for their in vivo melanocytotoxic effects in the zebrafish model. Accordingly, embryos were exposed to the different doses of ethanol extracts at the 6-hpf stage (when melanocytes are still not derived from the progenitor cells) and were compared in the melanocyte pattern (morphology, size, and distribution over the head and the yolk sac) with the embryos treated at the 24-hpf stage (when melanocytes had already been developed and started producing the pigment melanin).

The results of comparative analysis showed that neither LSE or ASE were melanocytotoxic, even if applied at high depigmenting doses of 400–500 µg/mL, since the embryos exposed from 6 hpf and 24 hpf had a similar melanocyte pattern (Figure 3), like those in the untreated group. Moreover, there was markedly less pigmentation in melanocytes upon LSE and ASE treatments than in the control group; this clearly indicates that the depigmenting effect of both extracts was achieved by inhibiting melanin biosynthesis. Similarly, kojic acid reduced melanocytes pigmentation at a dose of 2500 µg/mL without affecting the melanocytes number and morphology; this is in line with the results of previous in vitro and in vivo studies reporting that the activity of kojic acid is based on the inhibition of melanin synthesis, not on the melanocytotoxic activity [40]. On the other side, the melanocyte pattern in hydroquinone-treated embryos has largely depended on the dose and time when this agent was administered. In the 6-hpf treated group, the melanocytes were completely changed compared to that in DMSO-treated ones. However, the control embryos had large and stellate cells, and the melanocytes exposed to hydroquinone were greatly reduced in number and developed as small, round, or punctate, and were barely visible cells (Figure 3). This represents clear signs of melanocytotoxicity [41]. On the other side, hydroquinone applied to 24-hpf-old embryos exerted anti-melanogenic activity at doses < 10 µg/mL (embryos had stellate melanocytes) and melanocytotoxic effects at doses ≥ 10 µg/mL (melanocytes were punctate or non-visible) (Figure 3).

### 4.3. Fungal Extracts Are Not Inflammatory or Immunosuppressive upon Depigmenting Doses

Since the topical use of depigmenting agents may be associated with the skin inflammation or local immunosuppression, ASE and LSE were examined for the possible adverse effects on the neutrophils in vivo. Neutrophils are major innate immunity cells involved in skin inflammatory reactions and the primary immune response to microbial infection, including the skin as a first barrier. Several studies have demonstrated that, after topical application, hydroquinone is rapidly transported from the epidermis to the blood circulation, and may be hematotoxic to bone marrow [42], while effects on the circulating neutrophils may make them unresponsive against bacterial infection [43]. Therefore, the use of depigmenting agents without both inflammatory and immunosuppressive effects towards innate immune cells is of a particular importance in the treatments of skin hyperpigmentation disorders, especially in individuals with debilitated immunity, which are very susceptible to the microbial infections.

To address whether ASE and LSE are capable of provoking inflammation or neutropenia during anti-melanogenic treatments, we exposed transgenic *Tg*(mpx:GFP)i114 zebrafish embryos with fluorescently labeled neutrophils, enabling us to directly follow the effect of applied treatments on circulating neutrophils in real time. The *Tg*(mpx:GFP)i114 embryos were treated with ASE and LSE at the 24-hpf stage, when neutrophils are already differentiated and positive for the myeloperoxidase (Mpx) enzyme activity. Hydroquinone, known for the cytotoxic effect on neutrophils [44], was used as a control.

Obtained results showed that neither *A. silvaticus* or *L. sulphureus* are myelotoxic at depigmenting doses up to 400 and 500 µg/mL, respectively (Figure 4). On the other side, the treatments with kojic acid and hydroquinone led to the dose-dependent neutropenia in exposed embryos, demonstrating their immunosuppressive effect as reported in the literature. Taken together, data obtained in this study strongly indicate that the explored extracts with potent anti-melanogenic activity could be used in treatments of skin hyperpigmentation disorders.

## 5. Conclusions

Based on clinical issues of the existing skin-whitening agents and a great need for safe agents in the effective treatment of skin hyperpigmentation disorders, as well as the growing demand for natural skin care products, we explored the ethanol extracts of five edible mushrooms for anti-melanogenic potential and safety profile. Using the zebrafish embryo model as a preclinical animal platform, we identified the extracts of *A. silvaticus* and *L. sulphureus* with potent depigmenting activity, accompanied with a very good toxicity profile. The tested extracts successfully inhibited tyrosinase activity and melanin synthesis in zebrafish skin melanocytes, without melanocytotoxic, inflammatory, or immunosuppressive effects, which makes them valuable products for the possible topical application or the use as functional additives in cosmetics formulations.

Bearing in mind that mushroom extracts present the mixtures of chemically different compounds, their anti-melanogenic activity may be realized by targeting different molecular pathways involved in the process of melanogenesis (i.e., mitf; Tyr transcription factors; enzymes such as tyrosinase, adenylyl cyclase, dopachrome tautomerase, and DHICA oxidase; or melanosome trafficking). Therefore, further elucidation of chemical composition of *L. sulphureus* and *A. silvaticus* ethanol extracts that currently limit our knowledge on bioactive molecules responsible for extracts’ anti-melanogenic effects will enable us to address potency and mechanisms of action of the existing depigmenting compounds, as well as a spectrum of their other skin-beneficial properties, including anti-inflammatory and UV-protective activity. One of such worth-mentioning pluripotent and naturally derived molecules is melatonin, an ancient indolic derivative produced across diverse species of bacteria, fungi, plants, and animals, which exhibits multiple skin-beneficial properties, including anti-tyrosinase, antioxidative anti-inflammatory and anti-melanoma activities [45,46,47]. Therefore, additional efforts within our group will focus on isolation and the therapeutic potential of single melanogenesis-inhibiting compounds from the explored edible mushrooms.

## Figures and Tables

**Figure 1 jof-07-00834-f001:**
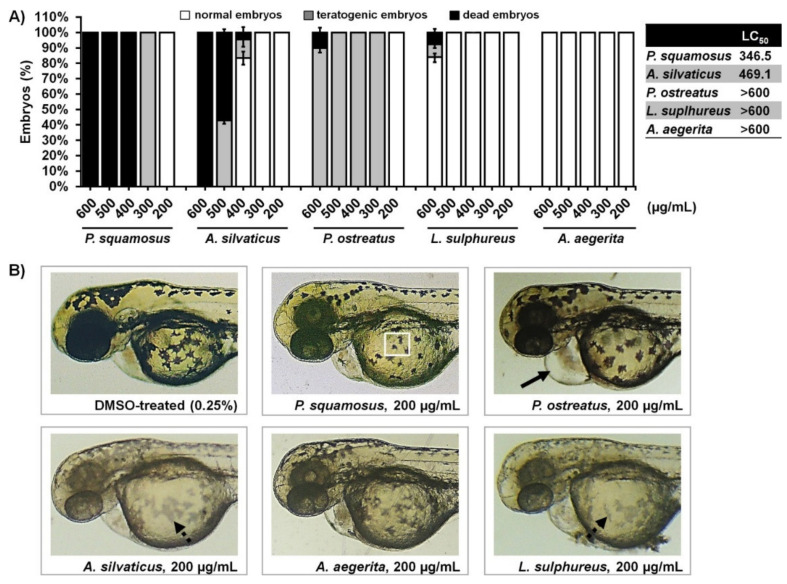
In vivo toxicity assessment of the ethanol extracts of five edible mushrooms in the zebrafish model, expressed as the lethal concentration (LC_50_). Wild-type (AB) embryos were exposed to the different doses of each extract (n = 60 embryos per dose) in a period from 6 hpf to 120 hpf, and inspected for (**A**) the survival and signs of teratogenicity, including cardiotoxicity, hepatotoxicity, and teratogenicity. (**B**) Morphology of the zebrafish embryos at 72 hpf stage being exposed to 200 µg/mL of each tested extracts is shown. At the applied dose, *P. ostreatus* caused a large pericardial edema (arrow), *P. squamosus* reduced size and morphology of the skin melanocytes (melanocytotoxicity) without inhibitory effect on cells pigmentation (boxed), *A. aegerita* was non-toxic, while *A. silvaticus* and *L. sulphureus* prevented the melanocytes pigmentation (dashed arrow) without the melanocytotoxic effect.

**Figure 2 jof-07-00834-f002:**
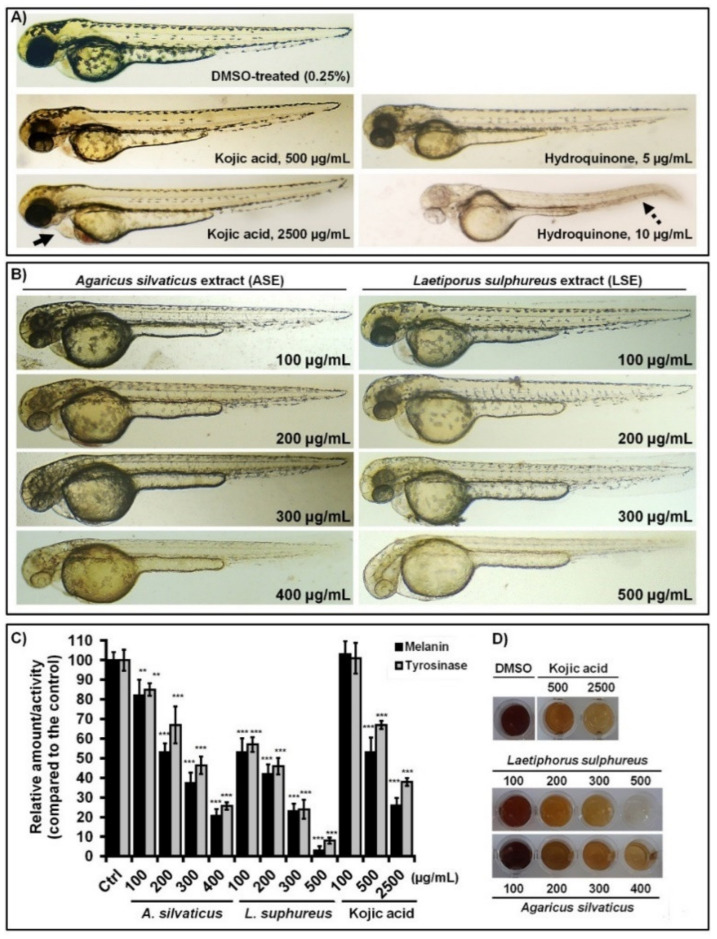
The ethanol extract of *A. silvaticus* and *L. sulphureus* successfully inhibited melanogenesis in zebrafish embryos without the adverse side effects on their development. Depigmenting efficacy of kojic and hydroquinone (**A**) and fungal extracts (**B**) assessed in embryos (n = 100) exposed to the different doses from 24 to 72 hpf are shown. Inhibitory effects of the tested extracts on the melanin synthesis and the tyrosinase activity at zebrafish embryos after 48 h-treatment (**C**) are shown, as well as on the melanin content in treated and untreated embryos (**D**). Statistically significant differences in the melanin content and tyrosinase activity between the control (DMSO-treated) and the treated groups (** *p* < 0.01; *** *p* < 0.001) are denoted.

**Figure 3 jof-07-00834-f003:**
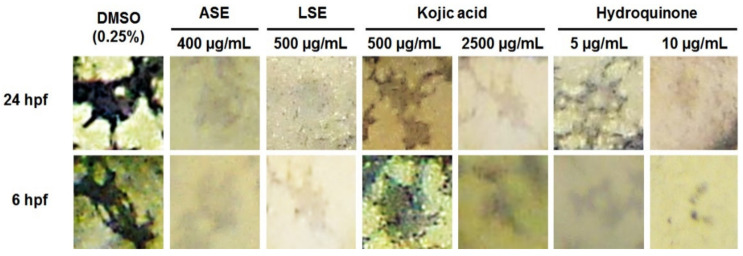
LSE and ASE are not toxic to melanocytes in vivo at the depigmenting doses. The inhibition of melanin synthesis and melanocytotoxicity were addressed in zebrafish embryos exposed to both extracts, i.e., kojic acid or hydroquinone at 6 hpf and 24 hpf. Melanocytes fading and non-changed morphology was observed in the treatments with ASE, LSE, and kojic acid, indicating the inhibition of melanin synthesis, while hydroquinone applied at 6 hpf exerted cytotoxic effects, manifested as the rounding of stellate melanocyte and the loss of their dendrites.

**Figure 4 jof-07-00834-f004:**
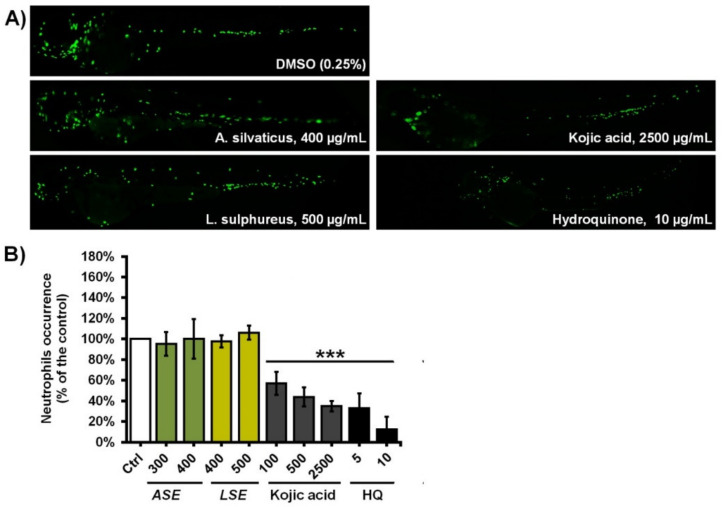
Depigmenting extracts ASE and LSE do not induce inflammation or neutropenia upon anti-melanogenic doses. (**A**) Morphology of *Tg*(*mpx*:GFP)i114 embryos exposed to various doses of ASE, LSE, kojic acid, and hydroquinone is shown. (**B**) The neutrophils’ occurrence and fluorescence intensity upon the different treatments is shown. Neither ASE or LSE provoked an inflammatory response or neutropenia in exposed embryos (n = 30), contrary to kojic acid and hydroquinone that were immunosuppressive. Statistically significant differences in the neutrophils’ occurrence between the control (DMSO-treated) and the treated groups (*** *p* < 0.001) are denoted.

**Table 1 jof-07-00834-t001:** Inhibitory potency of ASE, LSE, and kojic acid on the melanin synthesis and the tyrosinase activity assessed in the zebrafish embryos model.

Agent	IC_50_(Melanin Synthesis) (µg/mL) ^a^	IC_50_(Anti-Tyrosinase Activity) (µg/mL) ^b^
ASE	211.2	298.1
LSE	98.1	197.1
Kojic acid	1624.3	2082.6

^a^ IC_50_ value determined for the melanin synthesis present a concentration that decreased the melanin content for 50% in treated embryos compared to the untreated group. ^b^ IC_50_ value determined for the anti-tyrosinase activity presents a concentration leading to 50% inhibition of tyrosinase activity in treated embryos compared to untreated group.

## Data Availability

Not applicable.

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
