# Peer review of "Unravelling Anti-Melanogenic Potency of Edible Mushrooms Laetiporus sulphureus and Agaricus silvaticus In Vivo Using the Zebrafish Model"

_jof, 2021, doi:10.3390/jof7100834_

Round 1
Reviewer 1 Report
The topic is of interest and novel.
Methodology is also sound
My major critique is as follows
The authors use crude extracts and do not indentify the active compounds. Many analytical journals do not accept this. Therefore, I suggest to discuss this as limitation and provide reasonable explanation.
Why B16 melanoma was used but not normal human melanocytes? Again discuss limitations
Minor comments is that better overview of hormonal regulation of melanogenesis (Physiol Rev 84:1155-228, 2004) would be appreciated by the readers. Also mention melatonin as an melanin inhibiting factor (J Invest Dermatol 138:490-499, 2018)
Author Response
All authors of this manuscript are very thankful to both Reviewers and Editor for their comments, which greatly helped us to improve the quality our manuscript and better explain anti-melanogenic potential of the explored fungal extracts.
Reviewers’ comments and our response are presented below, and all changes within the manuscript are marked using the Track Changes option, as suggested
Responses to Reviewer 1 comments
Point 1: The authors use crude extracts and do not identify the active compounds. Many analytical journals do not accept this. Therefore, I suggest to discuss this as limitation and provide reasonable explanation.
Response 1: We are aware that chemical composition is an important trait in discovery of novel bioactive molecules, including those with anti-melanogenic properties. Therefore, the first step to accomplish this task is preliminary evaluating different extracts (made in various solvents differing in polarity at least) for depigmenting activity.
However, the presence of single bioactive compound or the certain group of active compounds (like total phenolic content) does not adequately reflect the entire biological potential of crude extract, since the mushroom extracts possess a large chemical diversity, which is much above single ingredients or mixture of a few of them. Aiming to chemically characterize different group of compounds with respect to their biological potential demands different procedures for preparation of the test sample, this may result in discrepancies in abundance/presence of compounds in two different extracts, thus losing meaning of the research. Moreover, as extracts are mixture of various compounds, it is often desirable to test the mixture as a whole, and not individual compounds per se. The detailed chemical analyses which focus on identification of all the present compounds in various mushroom extracts are a topic of its own and should be elaborated separately (commented with respect to these obtained results) in future.
Accordingly, the aim of our study was to evaluate depigmenting potential of crude extracts of various mushrooms, and select those with a possible topical use in the treatment of skin hyperpigmentations. Research interest of our two groups is focused on the application of natural extracts and products (preparations, formulations) in promoting human health (here on preventing skin hyperpigmentations), not on the use of single molecules, due to a few reasons:
1) Natural extracts are a mixture of chemically diverse compounds, where diverse compounds may possess depigmenting activity realizing it by targeting different molecular pathways involved in the process of melanogenesis (i.e. mitf, Tyr transcription factors, enzymes such as tyrosinase, adenylyl cyclase, dopachrome tautomerase, DHICA oxidase, or melanosome trafficking). This makes natural extracts as the effective mixtures in preventing an excessive skin melanogenesis.
2) In clinical practice and cosmetics, we are currently faced with a big issue regarding depigmenting molecules in sense that the compounds being both effective and safe are very rare; we learned this lesson with hydroquinone, kojic acid and many other molecules. Unfortunately, the majority of pure depigmenting agents are either effective but toxic/marginally toxic, or safe but weakly active. Moreover, current treatment options for pathological hyperpigmentation are marginally effective. One of very common drawback of pure depigmenting agents is their local immunotoxicity, manifested as the skin inflammation, irritation or local immunosuppression. In turn, the natural extracts as mixtures of bioactive compounds provide the opportunity to select ones with a potent anti-melanogenic activity but without immunotoxicity response in vivo. Moreover, providing a (bio)chemical diversity, natural extracts may simultaneously display depigmenting, anti-inflammatory and antimicrobial activity. This is main reason why we just explored the fungal extracts instead of single molecule(s) for depigmenting effect, safety and immunotoxicity, regarding them for possible topical application in form the extract-based cream, the extract-impregnated skin plasters, the extract-bearing biomaterials for topical use, and many other formulations.
Point 2: Why B16 melanoma was used but not normal human melanocytes? Again discuss limitations.
Response 2: Murine B16 melanoma cells are routinely and widely used in vitro model system in assessing the depigmenting potency of novel antimelanogenic compounds. Contrary to human A375 melanoma cells that weakly produced melanin, even upon α-MSH stimulation, B16 cells produce large amount of (especially in α-MSH presence), being presented in literature as a suitable in vitro model for anti-melanogenic studies and search for novel non-cytotoxic inhibitors of melanogenesis. Such one study has recently been published in Journal of Fungi (We et al., (2021) Investigations into Chemical Components from Monascus purpureus with Photoprotective and Anti-Melanogenic Activities. JoF, 7(8) 619),
However, having in the mind the crucial differences between in vitro and in vivo testing and a need to use a model system providing high similarity to humans, the major system employed in our study was the zebrafish model. This preclinical animal model was firstly used to address whether explored fungal extracts possess depigmenting activity, and secondly, to address are these extracts safe for the human use at effective depigmenting doses, with a particular attention on the skin melanocytotoxicity. As we found that L. sulphureus and A. silvaticus extracts are effective and safe at high doses of 400-500 µg/mL, B16 cells were only used as an additional model to confirm that extracts are not cytotoxic.
While B16 cells present the most commonly used in vitro model for the efficacy assessing of novel melanogenesis inhibitors, however, being aware of the contrasted response between melanoma cells (human and murine) and normal melanocytes (Flori et al., 2017) in proliferation upon α-MSH stimulation we have tested fungal extracts cytotoxic effect on B16 cells in absence of α-MSH, as done by We et al., (2021). Also, previous studies on melatonin showed that it was effective in preventing melanin synthesis both in B16 cells and in vivo (Velverde et al., 1995; Kim et al., 2014; Slomisnki et al., 2018).
We et al., (2021) Investigations into Chemical Components from Monascus purpureus with Photoprotective and Anti-Melanogenic Activities. JoF, 7(8) 619.
Flori et al., (2017) The α-melanocyte stimulating hormone/peroxisome proliferator activated receptor-γ pathway down-regulates proliferation in melanoma cell lines. Jpurnal of Experimental and Clinical Cancer Research, 36 (142).
Velverde P. et al. (1995) Melatonin antagonizes alpha-melanocyte-stimulating hormone enhancement of melanogenesis in mouse melanoma cells by blocking the hormone-induced accumulation of the c locus tyrosinase. Eur J Biochem 232(1):257-263.
Kim, T.-K. et al/ (2015) Melatonin and its metabolites accumulate in the human epidermis in vivo and inhibit proliferation and tyrosinase activity in epidermal melanocytes in vitro. Molecular and Cellular Endocrinology. 404, 1-8.
Slominski et al., (2018) Melatonin: A Cutaneous Perspective on Its Production, Metabolism, and Functions. Journal of Investigative Dermatology, 138(3):490-499
Point 3: Minor comments is that better overview of hormonal regulation of melanogenesis (Physiol Rev 84:1155-228, 2004) would be appreciated by the readers. Also mention melatonin as an melanin inhibiting factor (J Invest Dermatol 138:490-499, 2018)
Response 3: We are thankful to the reviewer suggestion and thus we added the recommended reference in the Introduction of the revised version of our manuscript, which describes the melatonin role in skin pigmentation process.

Reviewer 2 Report
The search for novel, effective and safe melanogenesis inhibitors is a field of growing interests in developing depigmenting agents to treat skin hyperpigmentations. The present study by Pavic et al. has shown that ethanol extract of edible mushrooms Laetiporus sulphureus and Agricus silvanticus effectively inhibited melanogenesis in a dose-dependent manner using the zebrafish model. Importantly, they did not exhibit toxicity and adverse effects at effective doses. The study is technically sound and the results are interesting. However, this article can be improved by addressing the following minor points.
Minor points:
1) Line 64-66: The authors referred to the difference between mushroom and human tyrosinase. In this regard, authors are advised to cite a paper or two on Thiamidol, a potent inhibitor of human tyrosinase but not mushroom tyrosinase (J Invest Dermatol 2018, 138, 1601 and 2019, 139, 1695).
2) Line 251-252: Check the grammar.
3) Figure 2 legend: The reviewer could not find explanation for C and D. Also, explanations for the arrows in A and B are missing here.
4) Table 1: 1624.28, 2082.6, and 197.14 should be 1624, 2083, and 197.1.
5) Line 166: The reviewer believes that 250 mg should be 250 ug.
6) Line 345: "the melanocytes was" should be "the melanocytes were".
7) Line 383: "hydroquinone leaded" should be "hydroquinone led".
8) Conclusions: The reviewer advices to add "Conclusions" at the end of Results and Discussion.
Author Response
All authors of this manuscript are very thankful to both Reviewers and Editor for their comments, which greatly helped us to improve the quality our manuscript and better explain anti-melanogenic potential of the explored fungal extracts.
Reviewers’ comments and our response are presented below, and all changes within the manuscript are marked using the Track Changes option, as suggested
Responses to Reviewer 2 comments
Point 1: Line 64-66: The authors referred to the difference between mushroom and human tyrosinase. In this regard, authors are advised to cite a paper or two on Thiamidol, a potent inhibitor of human tyrosinase but not mushroom tyrosinase (J Invest Dermatol 2018, 138, 1601 and 2019, 139, 1695).
Response 1: We are greatly thankful to the reviewer’s advice, and accordingly we have introduced two novel references related to thiamidol in the introduction of the revised manuscript version.
Point 2: Line 251-252: Check the grammar.
Response 2: The given sentence was reformulated as: Data on good safety profile of ASE and LSE obtained in the previous in vivo assay prompted us to investigate the depigmenting potency of these fungal extracts and thus possibility to be used as topical agents in treatment of skin hyperpigmentations.
Point 3: Figure 2 legend: The reviewer could not find explanation for C and D. Also, explanations for the arrows in A and B are missing here.
Response 3: The need explanation are included in the revised manuscript version.
Point 4: Table 1: 1624.28, 2082.6, and 197.14 should be 1624, 2083, and 197.1.
Response 4: According to the reviewer’s suggestion and data presented in the published papers in the Journal of Fungi we corrected the given values and presented them with one decimal.
Point 5: Line 166: The reviewer believes that 250 mg should be 250 ug.
Response 5: The given mistake regarding units has been corrected.
Point 6: Line 345: "the melanocytes was" should be "the melanocytes were".
Response 6: The given correction was introduced in the revised manuscript version.
Point 7: Line 383: "hydroquinone leaded" should be "hydroquinone led".
Response 7: The given correction was introduced in the revised manuscript version.
Point 8: Conclusions: The reviewer advices to add "Conclusions" at the end of Results and Discussion.
Response 8: We truly endorse the reviewer’s advice and help, and accordingly we added the recommended section at the end of our manuscript.

Round 2
Reviewer 1 Report
The effort to address the critique is appreciated. However the paper still needs revisions.
Instead of lengthy reply, which was not fully correct, the authors should made proper adjustment in the paper. For example, in clearly described limitation section to say that they do not know which factor(s) are responsible for phenotypic effects.
B16 model represents limitation. First, it is melanoma, second of murine origin. Please discuss limitations. Ideally experiments on normal human melanocytes should be performed. Their discussion of this topic in reply shows that they have limited expertise in melanin pigmentation, which is further enhanced by random reference list with many citations in low impact factor journals.
Than authors discuss melatonin in reply and, I do not see the referral to melatonin in the text. Please note that it is present across different species including plants, and it is also detected (with its metabolites) in natural products (Detection of Serotonin, Melatonin, and Their Metabolites in Honey. ACS Food Science & Technology 2021 1 (7), 1228-1235; DOI: 10.1021/acsfoodscitech.1c00119)
Author Response
All authors of this manuscript are very thankful to both Reviewer and Editor for their comments, which greatly helped us to improve the quality our manuscript and better explain anti-melanogenic potential of the explored fungal extracts.
Reviewers’ comments and our response are presented below, and all changes within the manuscript are marked using the Track Changes option, as suggested.
Responses to Reviewer 1 comments
Point 1: The effort to address the critique is appreciated. However the paper still needs revisions. Instead of lengthy reply, which was not fully correct, the authors should made proper adjustment in the paper. For example, in clearly described limitation section to say that they do not know which factor(s) are responsible for phenotypic effects.
Response 1: We agree with the reviewer that the lack of chemical analysis is limitation of this study. We have mentioned it in the section Conclusion, and pointed out that chemical analysis and assessment of depigmenting potential of single anti-melanogenic compounds will be a subject of our future study.
Point 2: B16 model represents limitation. First, it is melanoma, second of murine origin. Please discuss limitations. Ideally experiments on normal human melanocytes should be performed. Their discussion of this topic in reply shows that they have limited expertise in melanin pigmentation, which is further enhanced by random reference list with many citations in low impact factor journals.
Response 2: We completely agree with the reviewer’s comment that B16 cell are not the same as normal human melanocytes are, and that these murine melanoma cells do not completely reflect the biology of skin melanocytes in human. Unfortunately, there are still many papers being published every day in focused on the discovery of potent and non-toxic inhibitors melanogenesis by evaluating their activities on B16 melanoma cells. The reviewer is completely right when mentioned that the cytotoxicity testing on human melanocytes should be ideally experiment, but unfortunately, we are not capable to perform an additional experimental work. Instead of that, our research is a in vivo study, solely based on the zebrafish model as a preclinical animal model. Therefore, respecting the reviewer’s comment, we have decided to exclude the results on B16 cells, especially because this assay is of a marginal importance in relation to in vivo zebrafish-based experiments on which the whole study is based.
Point 3: Than authors discuss melatonin in reply and, I do not see the referral to melatonin in the text. Please note that it is present across different species including plants, and it is also detected (with its metabolites) in natural products (Detection of Serotonin, Melatonin, and Their Metabolites in Honey. ACS Food Science & Technology 2021 1 (7), 1228-1235; DOI: 10.1021/acsfoodscitech.1c00119)
Response 3: We are thankful to the reviewer’s efforts and suggestions related to melatonin. We specially paid attention on melatonin as an ancient natural molecule implicated in the melanogenesis regulation. Accordingly three new references, as suggested, have been included referencing on the studies related to melatonin origin, production and functions.